# Study on the Effect of Temperature on the Crystal Transformation of Microporous Calcium Silicate Synthesized of Extraction Silicon Solution from Fly Ash

**DOI:** 10.3390/ma16062154

**Published:** 2023-03-07

**Authors:** Dong Kang, Zhijie Yang, De Zhang, Yang Jiao, Chenyang Fang, Kaiyue Wang

**Affiliations:** 1School of Mining and Technology, Inner Mongolia University of Technology, Hohhot 010051, China; 2Key Laboratory for Green Development of Mineral Resources, Inner Mongolia University of Technology, Hohhot 010051, China; 3Inner Mongolia Engineering Research Center of Geological Technology and Geotechnical Engineering, Inner Mongolia University of Technology, Hohhot 010051, China; 4Key Laboratory of Geological Hazards and Geotechnical Engineering Defense in Sandy and Drought Regions at Universities of Inner Mongolia Autonomous Region, Inner Mongolia University of Technology, Hohhot 010051, China

**Keywords:** microporous calcium silicate, extraction silicon solution, crystal transformation, fly ash

## Abstract

In this study, microporous calcium silicate was synthesized from a silicon solution of fly ash extracted by soaking in strong alkali as a silicon source. By means of XRD, TEM, FTIR, and thermodynamic calculations, the crystal evolution and growth process of microporous calcium silicate were studied under the synthesis temperature of 295~365 K. The results show that calcium silicate is a single-chain structure of the Si–O tetrahedron: Q1 type Si–O tetrahedron is located at both ends of the chain, and the middle is the [SiO_4_^4−^] tetrahedron connected by [O^2−^] coplanar, and Ca^2+^ is embedded in the interlayer structure of calcium silicate. The formation rate and crystallization degree of calcium silicate hydrate were positively correlated with temperature. When the synthesis temperature was 295 K, its particle size was about 8 μm, and when the synthesis temperature was 330 K, a large number of amorphous microporous calcium silicate with a particle size of about 14 μm will be generated. When the temperature was above 350 K, the average particle size was about 17 μm. The microporous calcium silicate showed obvious crystalline characteristics, which indicate that the crystallization degree and particle size of microporous calcium silicate could be controlled by a reasonable synthesis temperature adjustment.

## 1. Introduction

Calcium silicate hydrate (C–S–H) is a general term for various ternary compounds composed of CaO–SiO_2_–H_2_O through different molar ratios. In the process of the high-value utilization [1,2,3,4,5,6] of fly ash [7,8,9,10], the amorphous silicon element is extracted by the alkali leaching method, and the obtained high-alkali extraction silicon solution (ESS) contains a large amount of silicon element. Microporous calcium silicate [11,12,13,14,15] is a kind of C–S–H with a special structure and is dynamically synthesized using amorphous silicon in ESS as a silicon source and adding a calcium source. It can be used as building materials [16,17,18], thermal insulation materials [19,20], papermaking filler, adsorbent, etc., so has a high utilization value. Many scholars such as Fang Qi and Lothenbach have studied the synthesis of microporous calcium silicate at temperatures ranging from 20 °C to 80 °C. The results show that microporous calcium silicate with low crystallinity is easy to generate at low temperatures, and its particle size gradually increases with the increase in the synthesis temperature [21]. Zhijie Yang [22] found that with the increase in the C/S molar ratio, the average pore volume, average pore diameter, and average specific surface area of the synthesized calcium silicate minerals decreased first, then increased and then decreased again. Grangeon [23] believed that the increase in the synthesis temperature would promote the agglomeration behavior of microporous calcium silicate and the formation of more dimers, thus leading to the increase in the synthetic particle size. Sonja Haastrup [14] synthesized microporous calcium silicate at 60~95 °C, and the experimental results showed that the synthesis of microporous calcium silicate with a short-range order in the structure had the characteristics of a long-range disorder. The increase in the calcium silicon ratio will lead to the shortening of the silicate chain, and the metal cation will be attached to the negatively charged hydroxyl group, resulting in the reduction in an available hydroxyl group in the synthesis of microporous calcium silicate. The increase in temperature will promote the dissolution of amorphous silicon, thus accelerating the synthesis process of microporous calcium silicate. S. Shaw et al. found that when the synthesis temperature increased to 190~310 °C, the synthetic products evolved from poorly crystallized microporous calcium silicate into synthetic products with a higher crystallized degree such as tobermorite and xonotlite-type calcium silicate crystals [24]. At present, the structure of calcium silicate synthesized under similar synthetic conditions is very different, and the microstructure, growth process, and crystal transformation law of calcium silicate are still not fully understood.

Temperature is a crucial influencing factor for the synthesis of calcium silicate minerals by ESS. Studying the influence of different synthesis temperatures on the generated calcium silicate is an extremely important way to master its crystal development and transformation process. In this paper, ESS was used as the silicon source and lime milk as the calcium source. In the range of synthesis temperature from 295 K (22 °C) to 365 K (92 °C), different synthesis temperatures were set as reaction variables to synthesize microporous calcium silicate. The synthetic products were analyzed by XRD, TEM, FTTR, and other modern analysis and testing techniques, and the thermodynamic reaction formula was established. The Gibbs free energies of various calcium silicate minerals in the synthetic products were calculated. Combining the experimental results with thermodynamic calculation, the formation sequence and possibility of various calcium silicate minerals synthesized were judged. The purpose was to synthesize microporous calcium silicate from the amorphous silicon of solid waste-fly ash, and to clarify the influence of temperature on the crystal transformation and growth process of microporous calcium silicate synthesized from ESS. Thus, the transformation process and development rule of microporous calcium silicate were mastered, which provided a scientific basis for the synthesis of microporous calcium silicate by ESS.

## 2. Materials and Methods

### 2.1. Materials

The fly ash (C = chemical composition is shown in Table 1) used in this test came from Datang International Power Generation Co. Ltd. in Inner Mongolia, China. NaOH (Kermel, China) was the analytical reagent for ESS preparation, distilled water was used as the experimental water, and a CaO (Kermel, China) analytical reagent was used as the calcium source for the synthesis of microporous calcium silicate.

### 2.2. Methods

#### 2.2.1. Preparation of ESS

ESS was prepared in a constant temperature water bath (SN-HWS-260, Shangyi, China). Under the condition of reaction temperature of 363 K (90 °C), the fly ash was soaked with 10% sodium hydroxide solution, the liquid–solid ratio was 4:1, and the reaction time was 90 min. Amorphous silicon reacts with NaOH in the fly ash to form a sodium silicate solution. In the form of Na_2_O·xSiO_2_ in the solution, and through circulation pump (SHZ-D Yuying, China) filtration, ESS can be obtained after solid and liquid separation, where the residual solid was used to extract aluminum and other valuable elements. The ESS components obtained in this experiment are shown in Table 2.

#### 2.2.2. Preparation of Microporous Calcium Silicate

Dissolve CaO in water to form a Ca(OH)_2_ solution with a concentration of 150g/L, and let stand for 30 min. After mixing ESS with lime milk at a silicon calcium molar ratio of 1:1, and adding distilled water to a solid–liquid ratio of 1:10, the synthesis experiment was carried out in a constant temperature water bath (Shangyi, China, SN-HWS-260) with a reaction time of 2 h. The microporous calcium silicate was synthesized at the following temperatures: 295 K (22 °C), 310 K (37 °C), 330 K (57 °C), 350 K (77 °C), 360 K (87 °C), and 365 K (92 °C). The filter cake obtained from the reaction products after vacuum filtration was dried at a drying temperature of 350 K for 24 h. The cake was manually ground with an agate mortar and passed through a 200-mesh sieve. As determined by XRF, the main chemical components of the synthesized calcium silicate are shown in Table 3, and the experimental process is shown in Figure 1. Finally, the microscopic morphology of microporous calcium silicate was observed by SEM (Hitachi S-4800, Japan), the structure of microporous calcium silicate was analyzed by XRD (PANalytical, The Netherlands, X’Pert Powder 3, Cu target, 40 kv, scanning range of 10~100°, step size 0.02°), FTIR (SHIMADZU, IRTracer-100, Japan) as well as TG and DSC (NETZSCH, Germany STA-449--F5 Heating rate of 5 °C/min), and the particle size of the synthesized samples at different temperatures was analyzed by a laser particle size analyzer (WJL-606, Jingke, China, refractive index 1.0).

## 3. Results and Discussion

### 3.1. Thermodynamic Analysis

Synthesis of hydrated calcium silicate is a very complex process. At present, more than 30 kinds of stable calcium silicate hydrate have been found. Under the same conditions of reaction time, silica–calcium ratio, alkali concentration, and stirring speed, calcium silicate products with different silica–calcium ratios were formed continuously. The calcium silicate minerals formed in the early stage also evolve continuously, thus forming some single or multiple composite calcium silicate products. In this paper, thermodynamic formulas were established from the perspective of thermodynamics to analyze the formation sequence and the possibility of various calcium silicate minerals under specific conditions. Equations (1) and (2) were used as the thermodynamic calculation formulas for each reactant and product. Using Equations (3)–(5) as the formulas for calculating the Gibbs free energy of various calcium silicate-like products:(1)$$△G_{T}=△H_{298}^{⦵}-T△S_{298}^{⦵}+\int_{298}^{T}△C_{p}dT-T\int_{298}^{T}\frac{△C_{p}}{T}dT$$
(2)$$△C_{p}=a+b*10^{-3}T+c*10^{-5}\;T^{-2}$$
(3)$$△H_{298}^{⦵}={\left(\Sigma △H_{298}^{⦵}\right)}_{resultant}-{\left(\Sigma △H_{298}^{⦵}\right)}_{reactant}$$
(4)$$△C_{p}={\left(\Sigma C_{p}\right)}_{resultant}-{\left(\Sigma △C_{p}\right)}_{reactant}$$
(5)$$△S_{298}^{⦵}={\left(\Sigma △S_{298}^{⦵}\right)}_{resultant}-{\left(\Sigma △S_{298}^{⦵}\right)}_{reactant}$$

The chemical reaction equations of various calcium silicate minerals in the synthetic products are shown as Equations (6)–(13):(6)$$3Ca{\left(OH\right)}_{2}\left(aq\right)+2Na_{2}SiO_{3}\left(aq\right)+2H_{2}O=3CaO\cdot 2SiO_{2}\cdot 3H_{2}O\left(s\right)+4NaOH\left(aq\right)$$
(7)$$Ca{\left(OH\right)}_{2}\left(aq\right)+2Na_{2}SiO_{3}\left(aq\right)+3H_{2}O=CaO\cdot 2SiO_{2}\cdot 2H_{2}O\left(s\right)+4NaOH\left(aq\right)$$
(8)$$2Ca{\left(OH\right)}_{2}\left(aq\right)+Na_{2}SiO_{3}\left(aq\right)+0.17H_{2}O=2CaO\cdot SiO_{2}\cdot 1.17H_{2}O\left(s\right)+2NaOH\left(aq\right)$$
(9)$$5Ca{\left(OH\right)}_{2}\left(aq\right)+6Na_{2}SiO_{3}\left(aq\right)+4H_{2}O=5CaO\cdot 6SiO_{2}\cdot 3H_{2}O\left(s\right)+12NaOH\left(aq\right)$$
(10)$$5Ca{\left(OH\right)}_{2}\left(aq\right)+6Na_{2}SiO_{3}\left(aq\right)+6.5H_{2}O=5CaO\cdot 6SiO_{2}\cdot 5.5H_{2}O\left(s\right)+12NaOH\left(aq\right)$$
(11)$$2Ca{\left(OH\right)}_{2}\left(aq\right)+3Na_{2}SiO_{3}\left(aq\right)+3.5H_{2}O=2CaO\cdot 3SiO_{2}\cdot 2.5H_{2}O\left(s\right)+6NaOH\left(aq\right)$$
(12)$$Ca{\left(OH\right)}_{2}\left(aq\right)+Na_{2}SiO_{3}\left(aq\right)+H_{2}O=CaO\cdot SiO_{2}\cdot H_{2}O\left(s\right)+2NaOH\left(aq\right)$$
(13)$$6Ca{\left(OH\right)}_{2}\left(aq\right)+6Na_{2}SiO_{3}\left(aq\right)+H_{2}O=6CaO\cdot 6SiO_{2}\cdot H_{2}O\left(s\right)+12NaOH\left(aq\right)$$

The related thermodynamic parameters of the reactants and products are shown in Table 4. Thermodynamic calculations were performed using HSC Chemistry (6.0), an integrated thermodynamic database software developed by the Outokumpu Research Center, Finland. The results were analyzed and plotted by Origin software (2021), as shown in Figure 2.

As shown in Figure 2, the thermodynamic calculation results show that temperature has an important influence on the growth process and crystal transformation of calcium silicate. At 275~350 K (37~77 °C), the Gibbs free energies for the synthesis of calcium silicate minerals with different Si/Ca ratios were all negative, which indicates that Equations (6)–(13) may all occur. Low temperature is conducive to the formation of calcium silicate hydrate containing more crystalline water, reaction Equations (9), (10), and (13) are more likely to occur than the others, that is, calcium silicate minerals with a silicon calcium ratio of 5:6 and 6:6 are more likely to be generated, which is consistent with the results of the XRD. With the gradual increase in the synthesis temperature, other C–S–H with a Si/Ca ratio gradually formed and transformed into the C–S–H crystal with a Si/Ca ratio of 5:6. With the continuous progress of the reaction, the xonotlite-type calcium silicate crystal with a Si/Ca ratio of 6:6 was finally formed. Calcium silicate minerals with different ratios, the degree of ease, or priority of formation will also vary with the reaction temperature, for example, 2CaO·3SiO_2_·2.5H_2_O is easier to generate than 2CaO·SiO_2_·1.17H_2_O when synthesized at room temperature. When the synthesis temperature is lower than 350 K, Equation (11) is more likely to occur than (8). In contrast, when the synthesis temperature was higher than 350 K, 2CaO·SiO_2_·1.17H_2_O is more likely to be generated. The relationship between the Gibbs free energy of various calcium silicates and temperature is shown in Table 5.

### 3.2. Phase Analysis

In order to determine the effect of different reaction temperatures on the synthesis of microporous calcium silicate, as shown in Figure 3, XRD analysis of the microporous calcium silicate samples synthesized at different reaction temperatures was carried out. XRD results showed that when the reaction temperature was higher than 295 K (22 °C) in the range of 310 K (37 °C)~365 K (92 °C), C–S–H was the main phase. When the reaction temperature was 295 K (22 °C), there was a diffraction peak of calcite at 29.40°, and there was no diffraction peak of calcium silicate minerals. When 2θ was 18.06°, 34.11°, 48.61°, and 54.38°, the characteristic diffraction peaks of Ca(OH)_2_ appeared. This shows that the sample still contained more residual Ca(OH)_2_, which was not involved in the synthesis reaction. With the increase in reaction temperature, the diffraction peak gradually sharpened and reached a higher 2θ, which means that the average base spacing of the microporous calcium silicate decreased.

As shown in Figure 3, when the reaction temperature was 295 K (22 °C), a large amount of Ca(OH)_2_ in the sample did not participate in the synthesis reaction, and there was no obvious diffraction peak of calcium silicate minerals in the XRD pattern. When the reaction temperature rose above 310 K (37 °C), most of the Ca(OH)_2_ in the raw material participated in the synthesis reaction, the diffraction peak of Ca(OH)_2_ gradually disappeared, and a small part of the calcium silicate with a silicon to calcium ratio of 5:6 was formed, but the diffraction fraction was diffuse and broad, and the crystallization condition was very poor, which is consistent with the thermodynamic theoretical calculation results, that when the reaction temperature reaches 350 K (77 °C) or above, the characteristic diffraction peaks of CaCO_3_ appear at 36.02°, 39.46°, 43.22°, and 47.63°. The residual Ca(OH)_2_ in the sample combined with CO_2_ in the air during drying to form calcium carbonate, and the diffraction peak of 2CaO·3SiO_2_·2.5H_2_O calcium silicate appeared at 29.49°, which confirms the previous thermodynamic theoretical calculation result. When the reaction temperature rose above 350 K (77 °C), the characteristic diffraction peaks of calcium silicate appeared at the 2θ positions of 29.42°, 32.25°, and 49.86°, respectively. The crystal characteristics of calcium silicate gradually became obvious, which was consistent with the TEM test results. When the reaction temperature was 360 K (87 °C), the diffraction peaks of CaO·2SiO_2_·2H_2_O and CaO·SiO_2_·H_2_O appeared at 32.32° and 49.76°. XRD analysis results combined with thermodynamic theoretical calculation showed that calcium silicate minerals with a silicon to calcium ratio of 5:6 were more readily produced than other types. With the progress of the synthesis reaction, calcium silicate products with different ratios of silicon to calcium were formed continuously, and all kinds of calcium silicate minerals formed in the early stage also transformed continuously, forming some single or multiple composite calcium silicate products. It is speculated that due to the relationship with reaction time, the formation of a xonotlite-type calcium silicate with a silicon calcium ratio of 6:6 could not be observed in the sample. The formation rate and crystallization degree of calcium silicate were positively correlated with temperature. With the continuous increase in reaction temperature, the diffraction peak of calcium silicate was significantly increased and enhanced, indicating that the amount of calcium silicate gradually increased, the peak value became gradually higher, and the peak type gradually steeper, which represents the gradual development of calcium silicate crystals. The diffraction peak of calcium silicate gradually shifted to a higher 2θ direction, indicating that the average substrate spacing of C–S–H decreased, and more calcium in the diffraction pattern was reflected in the XRD pattern in the form of CaCO_3_, indicating that the crystallinity of calcium silicate was still poor.

### 3.3. Microstructure Analysis

When the reaction temperature was 310 K (37 °C), as shown in the SEM (Figure 4b), the morphology of the microporous calcium silicate initially appeared in the reactants, but it was not obvious. When the reaction temperature was 330 K (57 °C), the generated calcium silicate began to form in large quantities. In Figure 4c, obvious micropores appeared in some areas, and the calcium silicate particles showed an obvious agglomeration behavior. At this time, the pore size of the micropores was about 0.3 μm. Compared with the XRD analysis, it can be seen that the micropore calcium silicate crystals generated at this time were poor, and the corresponding TEM (Figure 5a) showed that no crystal characteristics were observed in the reaction products at this time, The synthesis temperature was 350K (77 °C), as shown in Figure 4d, and the SEM diagram shows that the morphology of microporous calcium silicate appeared in more and more areas, and the honeycomb shape became more obvious. Combined with TEM (Figure 5b), the electron diffraction pattern showed a wide and diffuse diffraction ring, indicating that the microporous calcium silicate generated at this temperature still had a typical amorphous structure. With the increase in the reaction temperature, the amount of calcium silicate gradually increased, the crystal shape gradually developed, and the microporous morphology was more and more obvious. When the reaction temperature reached 360 K, the pore size increased to about 0.6–1 μm, as shown in Figure 4e, where its surface was porous and well-developed, and obvious diffraction spots appeared around the electron diffraction pattern ring, indicating that the microporous calcium silicate generated at this temperature had crystalline characteristics. In combination with the TEM (Figure 5c), it can be seen that obvious lattice patterns appeared during the synthesis of calcium silicate, indicating that its crystal form transformed from an irregular amorphous state to a regular crystal state. When the reaction temperature rose to 365 K, the morphology of the micropores showed a more obvious open petal. The results of the elemental analysis showed that the synthesized C–S–H was mainly composed of Si, Ca, O, and Na elements. In the high-alkali system, the Na element was the main impurity component in the microporous calcium silicate, which was consistent with the composition analysis in Table 4.

### 3.4. FTIR and TG–DSC Analysis

The infrared spectrum is shown in Figure 6. In the high wave-number region, an absorption peak at 3643.6 cm^−1^ could be observed, which was the –OH stretching vibration peak, corresponding to the –OH bending vibration at 1632.8 cm^−1^. It is speculated that the absorption peak is caused by the presence of a small amount of water in the sample and the presence of the O–H bond in the microporous calcium silicate sample. The bending vibration of the Si–O bond was at 664 cm^−1^ (Q2) and 870.9 cm^−1^ (Q1), and the stretching vibration of the Si–O bond was at 958.3 cm^−1^ (Q2) and 445.3 cm^−1^ (Q1), No Si–O (Q0), and Si–O (Q3) were found in the six samples. Therefore, it is speculated that the structure of the microporous calcium silicate synthesized is a Si–O tetrahedral single-chain, and each of the two adjacent Q2 [SiO_4_^4−^] tetrahedra is linked to the Si–O bond by the [O^2−^] co-top link, and Ca^2+^ is embedded in the interlayer structure of calcium silicate, while 1415.3 cm^−1^ corresponds to the stretching vibration of C–O.

The TG curves of the porous calcium silicate are shown in Figure 7, indicating that microporous calcium silicate had three stages of weight loss from room temperature to 1470 K. The first stage of weight loss occurred between 320 K and 570 K, and the weight loss rate was about 12–15%, which was mainly caused by a small amount of free water in the raw material and the removal of crystal water in the structure. In this temperature range, a large endothermic peak appeared on the differential scanning calorimetry curve (DSC curve) around 400 K corresponding to it. At this time, the crystal of the sample had not changed, and its main components were still amorphous. In the range of 570~1070 K, the weight loss of the second stage occurred, and the weight loss rate was about 5–7%, as a part of Ca(OH)_2_ in the sample absorbs CO_2_ in the air and turns into calcium carbonate, and the decomposition of CaCO_3_ and the residual Ca(OH)_2_ in the sample are endothermic decomposed into CaO and H_2_O. In this temperature range, the bond fracture of Si–OH resulted in the transformation of dehydrated microporous calcium silicate into calcium silicate without –OH(Ca_6_Si_6_O_18_). These three reasons together caused the TG curve in the range of 870~970 K to show obvious weight loss, while the dehydrated calcium silicate showed an endothermic reaction during dehydroxylation, leading to a relatively weak endothermic peak in the differential scanning calorimetry curve (DSC curve) at about 970 K. The thermogravimetric curve (TG) was relatively flat after 1070 K, indicating that there was no obvious weightlessness. The DSC curve showed a moderate exothermic peak at about 1370 K, which was due to the transformation of wollastonite in the sample into the high-temperature variant Ca_3_Si_3_O_9_.The DSC curve showed a moderate exothermic peak at about 1370 K, which was due to the transformation of the sample from wollastonite to the high temperature variant Ca_3_Si_3_O_9_, so no obvious weight loss occurred, and the final weight residue rate of the sample was between 76% and 81%.

### 3.5. Chemical Composition and Particle Size Analysis

The chemical composition of microporous calcium silicate synthesized at different synthesis temperatures is shown in Table 3. When the reaction temperature was 295 K (22 °C), the reaction was slow due to the low reaction temperature; as a result, the content of SiO_2_ in the XRF detection of the synthetic species was only 37.31%. As the reaction temperature gradually increased, when the reaction temperature reached 310 K (37 °C), the chemical composition of the synthesized calcium silicate tended to be stable, with CaO accounting for about 40~42% and SiO_2_ accounting for about 50%. The proportion of CaO was smaller than that of SiO_2_, which may be caused by the substitution of Al^3+^ for Si^4+^ into the silico tetrahedron to form the Al–O tetrahedron in the high-alkali system.

The composition analysis of the synthesized microporous calcium silicate at different temperatures showed that Na and Al are the main components of the synthesized microporous calcium silicate impurities. The composition of ESS shows that in the process of fly ash desilication, a small part of Al^3+^ enter into ESS. It is speculated that this part of Al^3+^ replaces silicon atoms in the silicon oxygen tetrahedron in the synthesis of calcium silicate products and participates in the synthesis reaction. The surface of microporous calcium silicate is negatively charged. According to the principle of electric neutrality, Ca^2+^ acts as a balance charge and will exist in the interlayer structure of the microporous calcium silicate, Na^+^, which is abundant in ESS, and can also play a similar role to Ca^2+^ in balancing charge, but its electrostatic effect is weaker than Ca^2+^, thus replacing a small part of Ca^2+^ into calcium silicate synthesis. There is an obvious competitive relationship between Ca^2+^ and Na^+^ in the microporous calcium silicate structure. As shown in Table 3, among the six microporous calcium silicate samples prepared, due to the low reaction temperature of sample no. 1, the composition of microporous calcium silicate in the sample was less, so the reference value was low. The proportion of Ca in the other five samples was in order from less to more with the synthesis temperature from low to high, while the proportion of Na showed an obvious opposite trend, which well proved that there was a competitive relationship in the structure of calcium silicate due to balancing the negative charge on the surface of the microporous calcium silicate.

The particle size of microporous calcium silicate has a very important influence on its application in papermaking, filtration, and so on. The particle size distribution of microporous calcium silicate generated by hydration at different temperatures is shown in Figure 8. The particle size of the synthesized microporous calcium silicate was between 5 μm and 20 μm, and the particle size frequency curve of the samples was an approximately normal distribution. The particle size of the synthesized microporous calcium silicate particles increased gradually with the increase in the synthesis temperature: the particle size was about 8 μm at 295 K (22 °C) and corresponded to 14 μm at 330 K (57 °C). As the synthesis temperature continued to rise, when the temperature reached more than 350 K (77 °C), the average particle size reached about 17 μm, indicating that the temperature had a significant impact on the particle size of the calcium silicate. As the average particle size increased with the increase in the synthesis temperature, a reasonable adjustment of the synthesis temperature can be used to control the particle size of the generated microporous calcium silicate.

### 3.6. Synthesis Process Analysis

According to the above experimental results and discussion, in the process of extracting the silicon element from fly ash, not only does the amorphous SiO_2_ from fly ash react with NaOH to generate Na_2_SiO_3_, but also a small amount of minerals containing Al_2_O_3_ react with NaOH to generate NaAlO_2_. The main components of the reaction solution are Na2SiO_3_ and unreacted NaOH as well as a small amount of NaAlO_2_ and other trace substances. As shown in Figure 9, in the dynamic synthesis of microporous calcium silicate with ESS and lime milk, first, the Si–O bond in SiO_2_ is broken, which leads to the increase in freedom of the SiO_2_ tetrahedron to form the H_2_SiO_4_^2−^ group. The Si–O tetrahedron contained in it was linked to the Si–O tetrahedron contained in another H_2_SiO_4_^2−^ by [O^2−^], formed by the Q1 form of the Si–O tetrahedron separated at both ends. Every two adjacent Q2 tetrahedrons by the [O^2−^] co-top link the Si–O tetrahedron single-chain polymerization structure, Ca^2+^ enters the interchain structure and is linked to the Si–O bond, thus in space forms a chain to chain interlayer structure.

## 4. Conclusions

In this work, the influence of ESS as a silicon source on the synthesis of microporous calcium silicate in a high-alkali system was studied by combining thermodynamic theoretical calculation with experimental results. The major conclusions of this paper are summarized as follows:(1)The XRD test results combined with thermodynamic calculation show that when ESS is used to synthesize microporous calcium silicate, calcium silicate products with different Si/Ca ratios are formed continuously with the progress of synthesis reaction. The crystalline forms of calcium silicate minerals synthesized in the early stage are also transformed, and 5CaO·6SiO_2_·5.5H_2_O and 5CaO·6SiO_2_·3H_2_O are more easily generated. When the synthesis temperature is 350 K (77 °C), 2CaO·3SiO_2_·2.5H_2_O will be generated. As the synthesis temperature continues to rise, 2CaO·SiO_2_·1.17H_2_O will be generated when it reaches 330 K (57 °C), and CaO·SiO_2_·H_2_O will be generated, but will eventually transform into the xonotlite-type calcium silicate crystal with a silicon to calcium ratio of 6:6, which is consistent with the thermodynamic calculation results.(2)The formation rate, crystallization degree, and particle size of the calcium silicate hydrate are positively correlated with temperature. The particle size of the calcium silicate hydrate was about 8 μm when the temperature was 295 K (22 °C). When the synthesis temperature was 330 K (57 °C), the particle size increased to about 14 μm. When the temperature reached above 350 K (77 °C), the average particle size reached about 17 μm, and the synthesis temperature was 360 K (87 °C). Therefore, the synthesized calcium silicate showed obvious crystal characteristics, which marks the transition from irregular amorphous to ordered crystalline.(3)When using ESS to synthesize calcium silicate minerals, Na and Al are the main components of the impurities. The XRF test results showed that the proportion of Ca and Na in the sample showed an obvious inverse relationship. Na^+^ will replace a small part of Ca^2+^ into the calcium silicate compound, forming a competitive relationship in the microporous calcium silicate structure. It is speculated that the reason is that both Na^+^ and Ca^2+^ play a role in balancing charge, while the electrostatic effect of Na^+^ is weaker than Ca^2+^.

## Figures and Tables

**Figure 1 materials-16-02154-f001:**
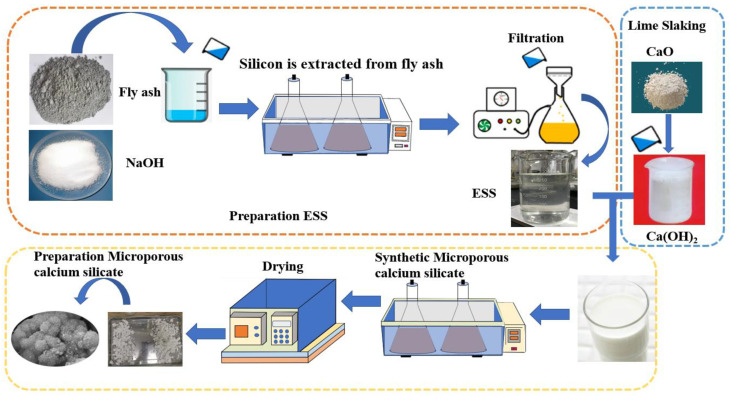
Preparation of microporous calcium silicate by the ESS form fly ash.

**Figure 2 materials-16-02154-f002:**
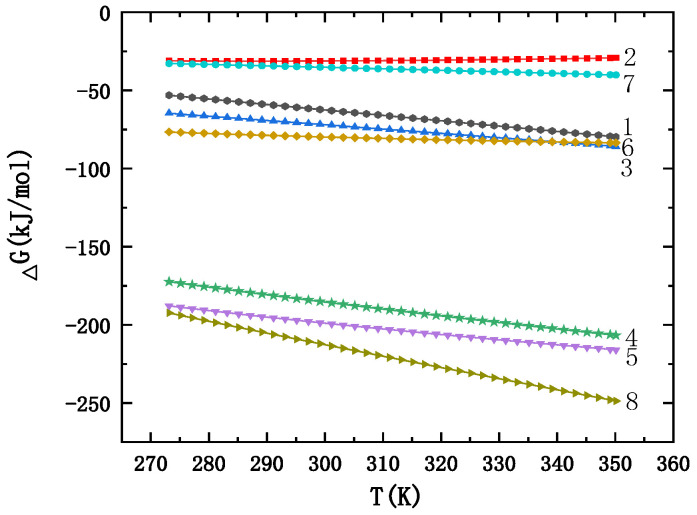
The relationship between the synthesis temperature and Gibbs free energy; 1—3CaO·2SiO_2_·3H_2_O; 2—CaO·2SiO_2_·2H_2_O; 3—2CaO·SiO_2_·1.17H_2_O; 4—5CaO·6SiO_2_·3H_2_O; 5—5CaO·6SiO_2_·5.5H_2_O, 6—2CaO·3SiO_2_·2.5H_2_O; 7—CaO·SiO_2_·H_2_O; 8—6CaO·6SiO_2_·H_2_O.

**Figure 3 materials-16-02154-f003:**
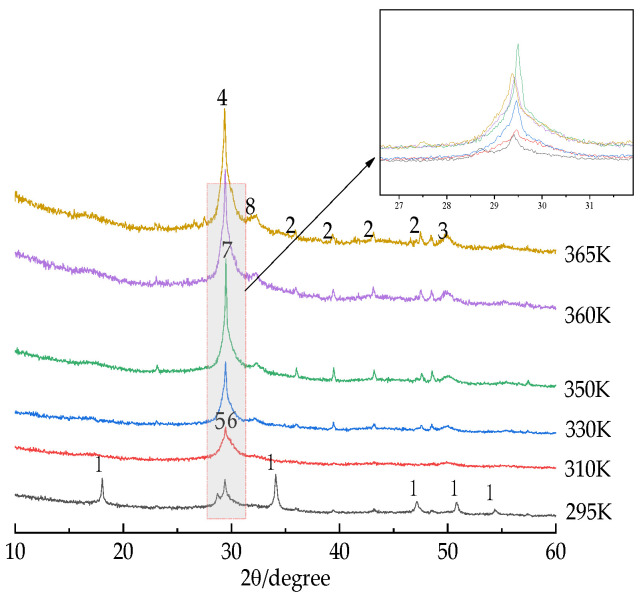
XRD patterns of the products at different temperatures. 1—Ca(OH)_2_; 2—CaCO_3_; 3—CaO·SiO_2_·H_2_O; 4—2CaO·SiO_2_·1.7H_2_O; 5—5CaO·6SiO_2_·5.5H_2_O; 6—5CaO·6SiO_2_·3H2O; 7—2CaO·3SiO_2_·2.5H_2_O; 8—CaO·2SiO_2_·2H_2_O.

**Figure 4 materials-16-02154-f004:**
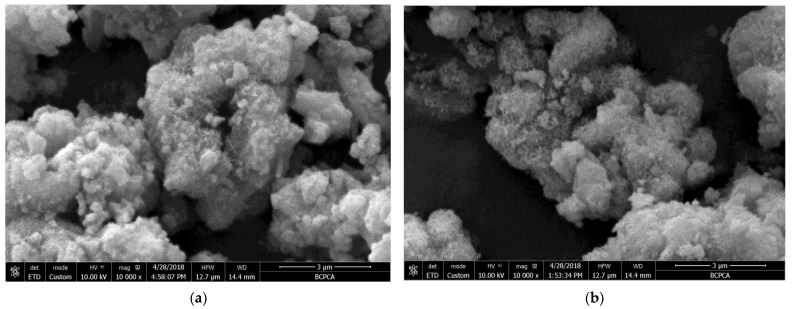
Micromorphology of the products at different temperatures: (**a**) 295 K; (**b**) 310 K; (**c**) 330 K; (**d**) 350 K; (**e**) 360 K; (**f**) 365 K.

**Figure 5 materials-16-02154-f005:**
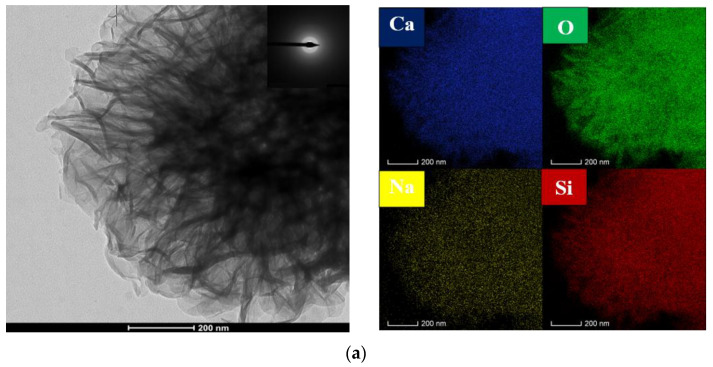
TEM of products at different temperatures: (**a**) 295 K; (**b**) 310 K; (**c**) 330 K.

**Figure 6 materials-16-02154-f006:**
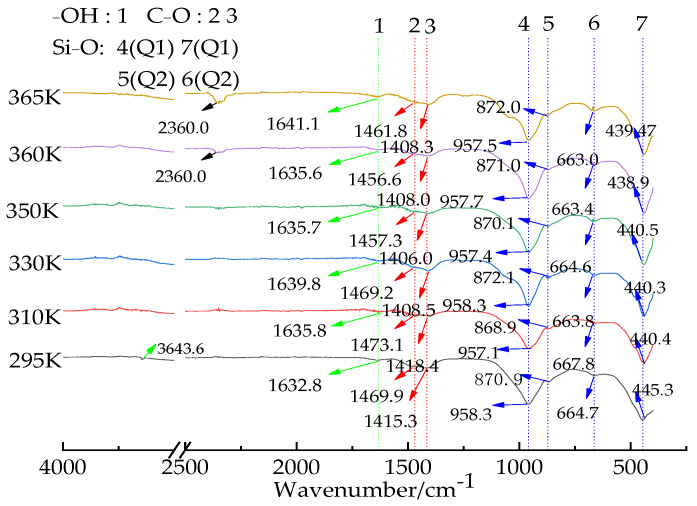
Infrared spectrum of the products at different temperatures.

**Figure 7 materials-16-02154-f007:**
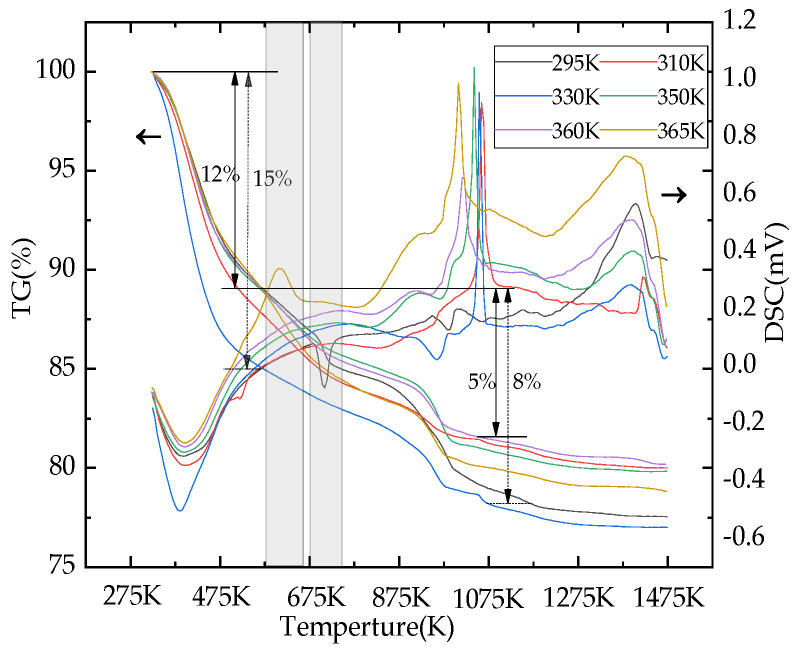
TG–DSC curves of the products at different temperatures.

**Figure 8 materials-16-02154-f008:**
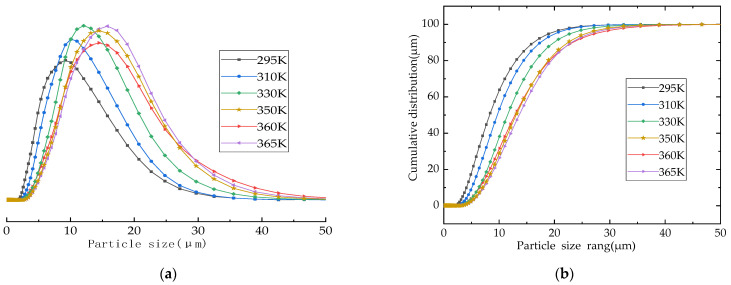
Particle size distribution at different temperatures: (**a**) particle size range; (**b**) cumulative distribution range.

**Figure 9 materials-16-02154-f009:**
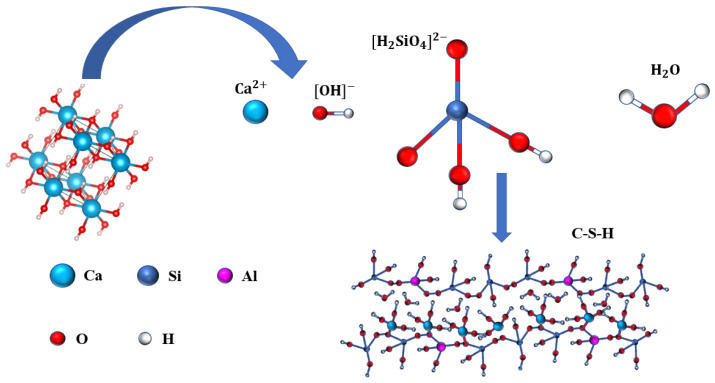
Molecular structure models of the microporous calcium silicate hydrate.

**Table 1 materials-16-02154-t001:** Chemical composition of fly ash.

Constituents	Content (g/L)
SiO_2_	47.9
Al_2_O_3_	40.1
CaO	4.09
MgO	0.65
Fe_2_O_3_	3.43
K_2_O	0.84
Na_2_O	0.32
SO_3_	0.52

**Table 2 materials-16-02154-t002:** Chemical composition of ESS.

Constituents	Content (g/L)
SiO_2_	40~60
NaOH	60~80
Al^3+^	0.5~2
Fe^3+^	0.01~0.05

**Table 3 materials-16-02154-t003:** The composition of microporous calcium silicate was synthesized at different temperatures.

Number	Temperature	Proportion (%)					
		CaO	SiO_2_	Al_2_O_3_	Na_2_O	MgO	Others
1	295 K	55.14	37.31	3.35	0.71	0.70	2.79
2	310 K	42.26	50.65	2.44	2.27	0.10	2.28
3	330 K	42.14	50.52	2.38	2.44	0.17	2.35
4	350 K	41.30	50.41	2.40	3.20	0.29	2.40
5	360 K	39.70	52.21	2.10	3.95	0.21	1.83
6	365 K	40.04	50.04	2.34	3.97	0.20	3.05

**Table 4 materials-16-02154-t004:** The related thermodynamic parameters of the products and reactants.

Chemical Formula	$△H_{298}^{⦵}\;(\mathrm{kJ}/\mathrm{mol})$	$△S_{298}^{⦵}\;(\mathrm{kJ}\;\times \;K)$	*a* + *b* × 10^−3^ + *c* × 10^−5^ T^−2^		
			*a*	*b*	*c*
3CaO·2SiO_2_·3H_2_O	−4782.312	312.126	341.163	188.698	−61.3790
CaO·2SiO_2_·2H_2_O	−3138.837	171.126	187.485	78.241	−43.304
2CaO·SiO_2_·1.17H_2_O	−2665.208	160.666	171.711	93.722	−30.962
5CaO·6SiO_2_·3H_2_O	−9935.745	513.168	600.613	312.545	−87.111
5CaO·6SiO_2_·5.5H_2_O	−10,686.770	611.492	462.750	791.194	0.000
2CaO·3SiO_2_·2.5H_2_O	−4920.384	271.542	332.503	151.879	−73.429
CaO·SiO_2_·H_2_O	−1917.845	112.707	263.112	64.952	−36.401
6CaO·6SiO_2_·H_2_O	−10,024.864	507.519	553.334	272.797	−76.776
Ca(OH)_2_	−1002.947	−74.517	89.248	33.150	−10.348
Na_2_SiO_3_	−1561.427	113.847	112.789	76.665	−19.708
H_2_O	−285.830	69.950	186.884	−464.247	−19.565
NaOH	−469.863	44.769	12,683.218	−52,451.899	−2228.366

**Table 5 materials-16-02154-t005:** The thermodynamic equations of various calcium silicates.

Number	Chemical Formula	The Equation for T(K) and ΔG(kJ/mol)
1	3CaO·2SiO_2_·3H_2_O	ΔG = −0.3455T + 41.1508
2	CaO·2SiO_2_·2H_2_O	ΔG =0.0257T − 38.66
3	2CaO·SiO_2_·1.17H_2_O	ΔG = −0.2754T + 10.6294
4	5CaO·6SiO_2_·3H_2_O	ΔG = −0.4426T − 52.4165
5	5CaO·6SiO_2_·5.5H_2_O	ΔG = −0.3622T − 89.9723
6	2CaO·3SiO_2_·2.5H_2_O	ΔG = −0.0891T − 52.8534
7	CaO·SiO_2_·H_2_O	ΔG = −0.0953T − 6.6816
8	6CaO·6SiO_2_·H_2_O	ΔG = −0.7291T + 6.1358

## Data Availability

The data presented in this study are available on request from the corresponding author. At the time the project was carried out, there was no obligation to make the data publicly available.

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
