# Peer review of "Study on the Effect of Temperature on the Crystal Transformation of Microporous Calcium Silicate Synthesized of Extraction Silicon Solution from Fly Ash"

_materials, 2023, doi:10.3390/ma16062154_

Round 1

Reviewer 1 Report

1. what is the novelty of the present study?

2. how the authors optimized the composition?

3. the methodology of preparation need to be elaborate?

4. What is the significance of the work

Reviewer 2 Report

Study on the effect of temperature on the crystal transformation of microporous calcium silicate synthesized of extraction silicon solution from fly ash

The authors need to improve the technical quality of the paper, which is insufficient to approve it for potential publication.

1.- The first critical issue throughout the manuscript is the vast English grammar errors. So, this manuscript must be sent to a professional English grammar correction office,

2.- Generally, the authors must review the manuscript's format carefully; there are many typos and punctuation at the beginning and end of several sentences. 

3.- The authors must review the introduction section carefully because a crucial mistake was detected. As claimed in the paragraph, Page 2, line 69: it is contradictory because the hydrothermal synthesis processing in references [26-30] was not applied to synthesize the calcium silicate. This information is detrimental to the quality and credibility of the research work presented here. Therefore, all the statements claiming the experiments conducted in this case were conducted under hydrothermal conditions must be deleted and specified as a simple chemical dissolution process. Hydrothermal experiments are conducted under controlled temperature and time conditions, leading to an autogenous inner pressure that plays an essential role in achieving solid dissolution. The internal pressure inside the glass containers is insufficient to generate hydrothermal conditions. The introduction section needs further detailed revision to provide correct and truthful fundamentals; please check it and rewrite it.

4.- Section 2 Experimental procedure. The fly ash chemical composition must be given in detail. I also include the reagent chemicals purity employed. A general explanation of the preparation of the ESS (line 98), Is 360 K is the only temperature chosen to perform this experimental part. A head in the next paragraph, five temperature conditions are mentioned. Why?? Authors should avoid confusion for potential readers. This section requires concision review and rewritten entirely. The characterization must be separated from the methods section 2.2.

Thermodynamic fundamentals are standard in the scientific community, so this information is out of the scope of this manuscript. Please, delete this information. The authors must focus on those chemical equilibriums associated with their experimental results to expose discussion statements in the manuscript. Include the state of the chemical precursors and products for chemical reactions (6 – 13), including subindex (s) or (aq).

Figure captions do not provide the correct information for the readers; these are tedious and confusing; correct all, and make them clear and descriptive.

-       How does the thermodynamic calculation explain the microporous calcium silicate growth process? It is not solely related to the stability of the chemical equilibrium to produce the solid.

-       Page 11, line 301, the final weight of the remaining 301 sample was between 76%, and 81%; the unit to describe weight is incorrect must be "mg," "g," or "kg." Correct this sentence.

-       What do you mean by this "3.6.Aynthesis process Analysis"? It needs further revision.

- Why the XRD results differ from the TEM images and diffraction analysis conducted by this technique. In the TEM figures, only 5c shows the presence of crystallinity in the sample, small inset in the left bottom corner. However, no evidence of atomic ordering is observed in the central picture in the background; why?

- The calcium silicate formation mechanism requires further revision, too, and currently is not supported by the experimental results.

In general, the manuscript is tidy; discussions are inferences to be provided; however, these are speculative without any support from the data. The authors need to provide specific result descriptions and discussion as well. Currently, the manuscript resembles a descriptive data result without logical and systematic development. I am afraid this paper is not suitable for publication in its current state. It deserves to emphasize that the technical quality of the manuscript is deficient. I encourage the authors to make their best effort to consider all the mandatory comments included in the present review. Too much time is required to perform all the corrections.

6.- Conclusions, all the statements provided do not indicate the relevance and novelty of the present results, and the arguments provided are skeptical and not supported by the experimental results. As for all the manuscripts, this section must carefully review and write again.

Reviewer 3 Report

In this study, the authors synthesized porous calcium silicate from the reaction of extracted sodium silicate and hydrated calcium hydroxide at different synthesis temperatures. The experimental method is not detailed in the manuscript. From the reviewer’s opinion, the main XRD peak claimed to be related to the silicate phase is suspected to be related to calcite due to the carbonation of hydrated lime, as indicated by XRD and thermal analyses. The authors should resolve this issue. The significance of the research should be highlighted! 

English

Careful revision for English!!!

Abstract

L19: should be from the extraction of silicon solution… it should be silicate solution!!

Actually, the sentence is confusing (avoid repetability! extraction, extracted!!) and should be revised.

L22-L26: excessively long sentence

L24: The silico-oxygen tetrahedron should be the silicon-oxygen tetrahedron.

L32-L33: should be: could be controlled by a reasonable synthesis temperature adjustment.

Materials and methods

L96: The method should be explained in detail!

L100: The authors did not describe how they separated the pure form of Na2O·xSiO2.

L105-L109: Punctuation mark!

L109: The equivalent temperature in Celsius should be indicated as well.

L113: Table 4 becomes Table 1.

Figure 1 should come after L120.

L182: 2θ of 29.40° . Actually, this peak position is characteristic of calcite, as other minor characteristic peaks seem to appear in Figure 3 (2θ values of 24.90°, 27.04°, 32.82° and 29.40°). This indicates that your hydrated lime has been carbonated. Thermal analysis (TGA or DSC) will usually confirm that what you are claiming is silicate phase rather than calcite. Please check L276. The equivalent temperature in Celsius should be indicated as well.

Figure 7 should come after L302.

Figure 9 should be referenced if it does not belong to the authors.

L366: Is this a report?

Reviewer 4 Report

Excluding remarks to the English and the typos, I have the following comments and critical remarks on the manuscript:

1)      Many sentences of the text are inconsistent. For example, lines 114-120 or 191-204 are too long phrases and seem to be completely inconsistent. It is necessary to check and correct English and syntax. Commas instead of dots!?

2)      The experimental conditions for analyzing the material are described rather uninformatively. What means “and other modern testing techniques”? In the Methods section, all techniques should be named, and why and for what they were used should be indicated. Moreover, the manuscript contains insufficient data on sample preparation and data collection conditions. Just listing the devices is not enough.

3)      Figure 3 is badly designed. It is better to write formulas for phases in the caption to the figure, otherwise they are unreadable. Designations 2 and 3 are better transferred for the pattern at 350 K.

4)      Line 200. It's completely unclear how calcium can enter into the interlayer structure of the Si-O tetrahedron? How do the authors imagine it? This is an incorrect phrase, or a more detailed description of this process is required. But this is impossible.

5)      Line 271-272. Incorrect phrase.

6)      Line 275. Is the presence of the banding vibration of CO3 revealed?

7)      Line 278. FT-IR spectra? Missing word. Line 280 is the same. What exactly, spectrum or what is it?

8)      Line 349. How was the structure model established? This is not described in any way in the article.

 I strongly suggest that the authors of the present manuscript focus on the improvements. The English requires a brush up. The presentation of the results is written in a very poor English: they use well-known scientific words but in the wrong context. The manuscript must be rewritten under the supervision of a person fluent in English.

After improvements, the authors should try again to submit this material for consideration, however, in the current form, this manuscript should be rejected.

Round 2

Reviewer 2 Report

The authors fully corrected all the comments, and the technical quality of the manuscript was improved sufficiently.

Reviewer 3 Report

L310:Infrared spectrum is shown in Figure 6.

L312: Corresponding to

This work is still in need of major revision! Examples are given below:

L310-L314: Too long sentence.

L345-L348: long sentence

L358: as a result,

L394:17μm (space): 17 μm

Reviewer 4 Report

My comments were taken into account to a sufficient extent.
